# Data collected using small uncrewed aircraft system during the TRacking Aerosol Convection Interactions ExpeRiment (TRACER)

Francesca Lappin[1,6], Gijs de Boer[2,3,4], Petra Klein[6], Jonathan Hamilton[2,3], Michelle Spencer[1,6,7], Radiance Calmer[2,4], Antonio R. Segales[1], Michael Rhodes[4], Tyler M. Bell[1], Justin Buchli[4], Kelsey Britt[1,6,7], Elizabeth Asher[5], Isaac Medina[6], Brian Butterworth[2,3], Leia Otterstatter[6], Madison Ritsch[4], Bryony Puxley[6], Angelina Miller[4], Arianna Jordan[6], Ceu Gomez-Faulk[4], Elizabeth Smith[7], Steven Borenstein[2,5], Troy Thornberry[8], Brian Argrow[4], and Elizabeth Pillar-Little[1*]

[1]Cooperative Institute for Severe and High-Impact Weather Research and Operations, Norman, Oklahoma, USA
[2]Cooperative Institute for Research in Environmental Sciences, University of Colorado Boulder, Boulder, Colorado, USA
[3]NOAA Physical Sciences Laboratory, Boulder, Colorado, USA
[4]Integrated Remote and In Situ Sensing, University of Colorado Boulder, Boulder, Colorado, USA
[5]NOAA Global Monitoring Laboratory, Boulder, Colorado, USA
[6]University of Oklahoma, Norman, Oklahoma, USA
[7]NOAA National Severe Storms Laboratory, Norman, Oklahoma, USA
[8]NOAA Chemical Sciences Laboratory, Boulder, Colorado, USA
[*]Former affiliation

**Correspondence:** Francesca Lappin (francesca.lappin@ou.edu)

**Abstract.** The main goal of the TRacking Aerosol Convection interactions ExpeRiment (TRACER) project was to further understand the role that regional circulations and aerosol loading play in the convective cloud life cycle across the greater Houston, Texas area. To accomplish this goal, the United States Department of Energy and research partners collaborated to deploy atmospheric observing systems across the region. Cloud and precipitation radars, radiosondes, and air quality sensors captured atmospheric and cloud characteristics. A dense lower atmospheric dataset was developed using ground-based remote sensors, a tethersonde, and uncrewed aerial systems (UAS). TRACER-UAS is a subproject that deployed two UAS platforms to gather high-resolution observations in the lower atmosphere between 1 June and 30 September 2022. The University of Oklahoma CopterSonde and the University of Colorado Boulder RAAVEN (Robust Autonomous Aerial Vehicle – Endurant Nimble) were flown at two coastal locations between the Gulf of Mexico and Houston. The University of Colorado RAAVEN gathered measurements of atmospheric thermodynamic state, winds and turbulence, and aerosol size distribution. Meanwhile, the University of Oklahoma CopterSonde system operated on a regular basis to resolve the vertical structure of the thermodynamic and kinematic state. Together, a complementary dataset of over 200 flight hours across 61 days was generated, and data from each platform proved to be in strong agreement. In this paper, the platforms and respective data collection and processing are described. The dataset described herein provides information on boundary layer evolution, the sea breeze circulation, conditions prior to and nearby deep convection, and the vertical structure and evolution of aerosols.

# 1 Introduction

The Houston-Galveston region is a coastal metropolis where urbanization and industrialization have redefined the landscape and atmospheric composition. To accommodate the demand for space and energy, natural landscapes have been replaced by sprawling urban dwellings and industrial areas. These changes in land use affect surface roughness and fluxes, thereby impacting the structure and stability of the atmospheric boundary layer (ABL). All along the Gulf Coast, the sea breeze circulation (SBC) can trigger or enhance deep convection, leading to potentially heavy rains, which are exacerbated when natural drainage plains have been built over and trigger flash flooding events. Moreover, petrochemical plants dot the coastline, emitting aerosols and gaseous pollutants into the ABL, leading to high ozone and particulate matter events that are closely linked with the SBC (Caicedo et al., 2019; Li et al., 2020; Park et al., 2020). Although, the influence of aerosol loading on convection is still disputed in the literature, with some suggesting it enhances convection (Fan et al., 2020; Zhang et al., 2021) and others finding it inhibits convection (Grant and van den Heever, 2014; Varble, 2018; Park et al., 2020). The combination of these factors makes the Houston area vulnerable to hazardous air quality and heavy rain events, thus driving a need for enhanced observations to understand the processes accompanying a changing climate and landscape (Hagos et al., 2016).

The main goal of the TRacking Aerosol Convection interactions ExpeRiment (TRACER) is to further the understanding of the convective cloud lifecycle and its interplay with aerosols. This spans from shallow cloud modeling, which represents one of the great uncertainties in climate projections (Bony et al., 2015; Zhao et al., 2016), to deep convection and the intertwining of microphysical and dynamic processes that dictate storm intensity (Khain et al., 2005). Employing a suite of radars, air quality instruments, and surface flux stations, the campaign gathered observations to validate models and improve the physical parameterizations. While clouds form near the top of the ABL, their lifecycle begins within the ABL and depends on low-level moisture, composition, and momentum. The coastal Houston region creates a unique urban-coastal boundary layer with impacts from the urban heat island as well as the SBC. Thus, to fully understand the cloud lifecycle, dense observations of the ABL across the region are necessary. To develop an ABL dataset, ground-based remote sensors, radar wind profilers, and small uncrewed aerial systems (sUAS) were deployed. The subproject TRACER-UAS is the focus of this paper and utilized a fixed-wing and rotary-wing sUAS to gather high spatiotemporal observations of the ABL throughout four intensive observation periods (IOPs) during the main TRACER campaign.

Over the past decade, the use of UAS in weather research has expanded due to its proven utility across a range of meteorological conditions (Elston et al., 2011; Cassano, 2014; Elston et al., 2015; Båserud et al., 2016; Cione et al., 2016). Reineman et al. (2013, 2016) used a fixed-wing platform to gather turbulence measurements within the marine boundary layer. Flagg et al. (2018) found that assimilating data from the same platform improved the model's representation of the marine boundary layer and reduced bias. With careful consideration of sensor placement, UAS-collected data are of comparable quality to meteorological towers, radiosondes, and ground-based remote sensors (Barbieri et al., 2019; Bell et al., 2020). Observations from fixed-wing UAS deliver horizontal transects of the environmental state, capturing heterogeneities in the ABL during turbulent times, such as during the passage of the sea breeze front (SBF) or prior to convection initiation (CI). Rotary-wing UAS, which collect repeated vertical profiles, resolve the structural evolution of the ABL in transitional and pre-convection condi-

tions (de Boer et al., 2020; Lappin et al., 2022). The TRACER-UAS campaign deployed the University of Colorado Integrated Remote and In Situ Sensing (IRISS) Robust Autonomous Aerial Vehicle – Endurant Nimble (RAAVEN) and the University of Oklahoma CopterSonde UAS (Segales, 2022). Utilizing both platforms allows for a complex four-dimensional dataset of the ABL in complex terrain.

Understanding the context of processes in the ABL is necessary to fully understand the convective cloud lifecycle. Transportation of moisture, momentum, heat, and aerosols depends on the structure of the ABL and its relationship with the SBC. The complex interaction between this advection and mechanical lifting primes the pre-convective environment, but access to positively buoyant air is required to stimulate convection (Fovell, 2005; Hartigan et al., 2021; Fu et al., 2022). Although the SBC is often treated as steady state and homogeneous along the coastline, Puygrenier et al. (2005) found that the SBF pulsates due to convectively redistributing heat and weakening the pressure gradient force. Within the SBF, regions of enhanced convergence form due to the collision with horizontal convective rolls and increase vertical motion (Atkins et al., 1995; Iwai et al., 2008). Heterogeneities in the SBF are compounded by the addition of the bay breeze and urban land use (Miller et al., 2003; Chen et al., 2019). The SB can impact air quality in multiple ways, aerosols can be trapped near the surface or fumigated aloft depending on ABL stability (Verma et al., 2006; Iwai et al., 2008; Finardi et al., 2018). High ozone air masses can recirculate and increase the ozone residence time due to stagnation (Banta et al., 2005; Chen et al., 2011; Caicedo et al., 2019). Regional air quality is highly dependent on ABL characteristics, and much of the knowledge gathered on these processes has stemmed from models, even though the ABL representation in numerical modeling is overly simplified by parameterizations. Moreover, datasets to verify or improve model performance are lacking in nonhomogeneous boundary layers. In short, the need for a dense dataset of ABL observations is seen across many factions of atmospheric science, especially in complex terrain such as coastal cities.

The TRACER-UAS dataset encompasses the thermodynamic and kinematic data necessary to understand the structure, stability, and flux magnitude to interpret the ABL evolution in heterogeneous terrain. During the campaign, data were collected during sea breeze events, prior or near convection, and quiescent periods. In total, the CopterSonde and the RAAVEN collected over 200 flight hours worth of data at two flight sites. The two flight sites have differences in roughness length, air quality, and forcings which illustrate ABL heterogeneity in the region. The following paper will first describe each platform and the data processing more thoroughly. Subsequently, an overview of the conditions sampled and data comparison between both platforms is provided.

## 2 Description of Vehicles and Sensors

The TRACER-UAS project saw the deployment of two different sUAS, including the University of Colorado RAAVEN and the University of Oklahoma CopterSonde. These systems have been used extensively to collect atmospheric measurements in connection with several different field campaigns (e.g., de Boer et al., 2022; Cleary et al., 2022). As part of the preparations for TRACER, significant time was spent comparing measurements from the two platforms to radiosonde and tower-based measurements collected at the US Department of Energy (DOE) Atmospheric Radiation Measurement (ARM) Southern Great

Plains (SGP) facility. These efforts revealed that both platforms captured the state of the atmosphere with significant accuracy and were comparable to each other and to the ARM instrumentation. Additional details on this intercomparison can be found in de Boer et al. (2023). Below we provide an overview of each platform, along with additional references which provide detailed information on the sensor and system specifications.

## 2.1 University of Colorado RAAVEN

The University of Colorado Boulder RAAVEN (Fig. 1) is a fixed-wing sUAS that has been developed for the collection of detailed information on the structure of the atmosphere, and has been operated by the University of Colorado team since 2019. The RAAVEN airframe is based on the commercially-available DRAK UAS manufactured by RiteWing RC, and has a wingspan of 2.3 m. The airframe has been updated to meet the needs of atmospheric science missions spanning a variety of environments. The RAAVEN uses the PixHawk2 flight controller and is powered by an 8S 21000 mAh Lithium Ion (Li-Ion) battery pack to offer flight times around 2.5 hours with minimal payload. The airframe was modified to include a tail boom in order to assist with improvement of longitudinal stability and overall performance. The aircraft has a maximum airspeed of approximately 36 m s$^{-1}$, though during TRACER flights were generally conducted in the 16-19 m s$^{-1}$ range.

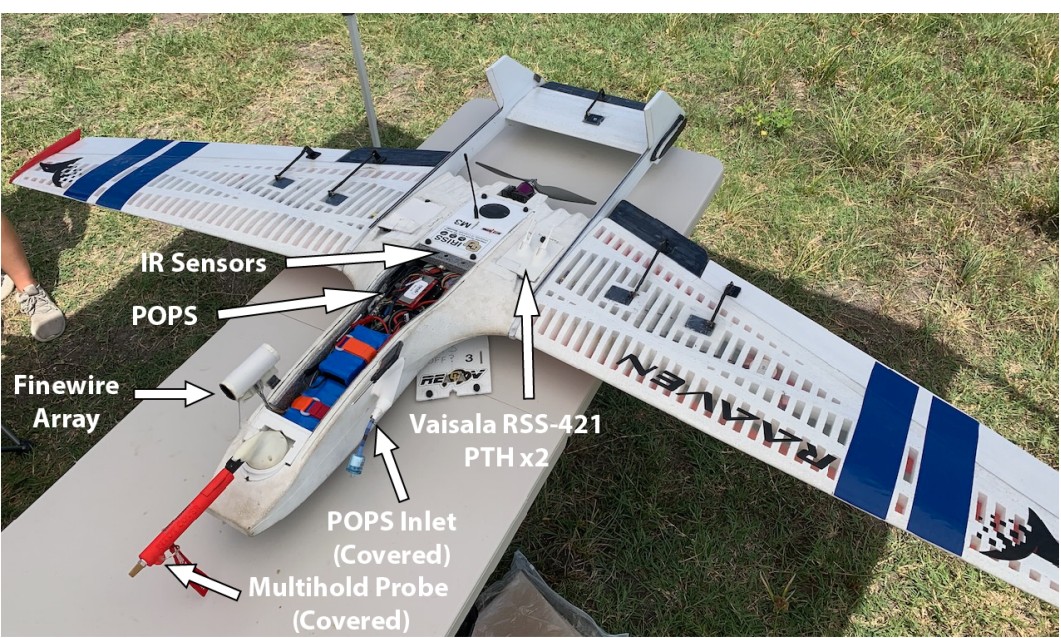

**Figure 1.** The RAAVEN UAS, as instrumented in the field for TRACER.

For TRACER, the RAAVEN carried sensors from the miniFlux payload co-developed by the National Oceanic and Atmospheric Administration (NOAA), the Cooperative Institute for Research in Environmental Sciences (CIRES) and Integrated Remote and In Situ Sensing (IRISS) at the University of Colorado. This features a primary suite of instruments (see Fig. 1), specifically including a pair of RSS421 PTH (pressure, temperature, humidity) sensors from Vaisala, Inc., a multihole pressure

probe (MHP) from Black Swift Technologies, LLC (BST), a pair of Melexis MLX90614 IR thermometers, a custom finewire array, developed and manufactured at the University of Colorado Boulder, and a VectorNav VN-300 inertial navigation system (INS).

In addition to the sensors described above, the RAAVEN carried a Printed Optical Particle Spectrometer (POPS, Gao et al. (2016)), developed by the NOAA Chemical Sciences Laboratory, and currently sold commercially by Handix Scientific. The POPS provides information on aerosol size distribution, giving size-resolved concentrations across 24 size bins, as well as information on a variety of instrument monitoring systems. POPS data are available for all but four of the completed flights, during which sensor overheating resulted in no usable data. This overheating issue was resolved by adding additional ventilation to the payload bay carrying the POPS sensor once it was discovered. POPS data quality has been evaluated in previous studies (e.g., Mei et al., 2020). The inlet for the POPS instrument was located on the RAAVEN fuselage and consisted of a 2 mm ID (3 mm OD) brass tube mounted in an isoaxial configuration. The POPS was operated at a sample flow rate of 3 cm$^3$ s$^{-1}$, resulting in sub-isokinetic sampling at RAAVEN airspeeds and leading to size dependent oversampling of particles. The sample line between the inlet and the instrument was constructed of brass and conductive silicone tubing. Overall particle sampling efficiency accounting for aspiration and transmission (Baron and Willeke, 2001) ranged from 1.1 to 2.1 over the POPS measurement size range. Data quality can additionally be impacted by a variety of different things, including sensor temperature and flow rate. These issues are diagnosed in post-processing, and the POPS data quality flag (see below) helps provide users with additional information on sensor data quality.

With this combination of sensors, the RAAVEN was configured to observe the atmospheric and surface properties necessary for evaluating kinematic and thermodynamic states, turbulent fluxes of heat and momentum, and the aerosol size distribution for particles between 150-2500 nm. The addition of this sensor suite reduced the aircraft endurance to approximately 90 minutes, as the POPS installation took up some of the space normally allocated for batteries. All sensors along with aircraft autopilot data were logged using a custom-designed FlexLogger data logging system. Detailed information on the performance of the different sensors and data acquisition rates can be found in de Boer et al. (2022) and Cleary et al. (2022) and are therefore not repeated here.

## 2.2 University of Oklahoma CopterSonde

The CopterSonde is a rotary-wing quadcopter used to collect frequent vertical profiles of the ABL (Fig. 2). It was developed at the University of Oklahoma and is maintained by Cooperative Institute for Severe and High-Impact Weather Research and Operations (CIWRO). The platform is 0.5 m in diameter and weighs 2.3 kg, making it easily transportable. The CopterSonde uses a combination of direct sensors, autopilot software, and algorithms to gather a profile of atmospheric data. Pressure, temperature, and humidity are observed using an MS5611 barometric pressure sensor, iMet-XF bead thermistor, and HYT-271 humidity capacitor, respectively. The pressure sensor is integrated into the Pixhawk CubeOrange autopilot board to improve the altimeter estimation. To remove temperature fluctuations from the pressure observations, the Pixhawk is heated to a constant temperature within the first two minutes of start-up. The thermistors and humidity capacitors are housed in the intake scoop of the CopterSonde, where they are sheltered from insolation and heat from the motors or Pixhawk Cube. The positioning of

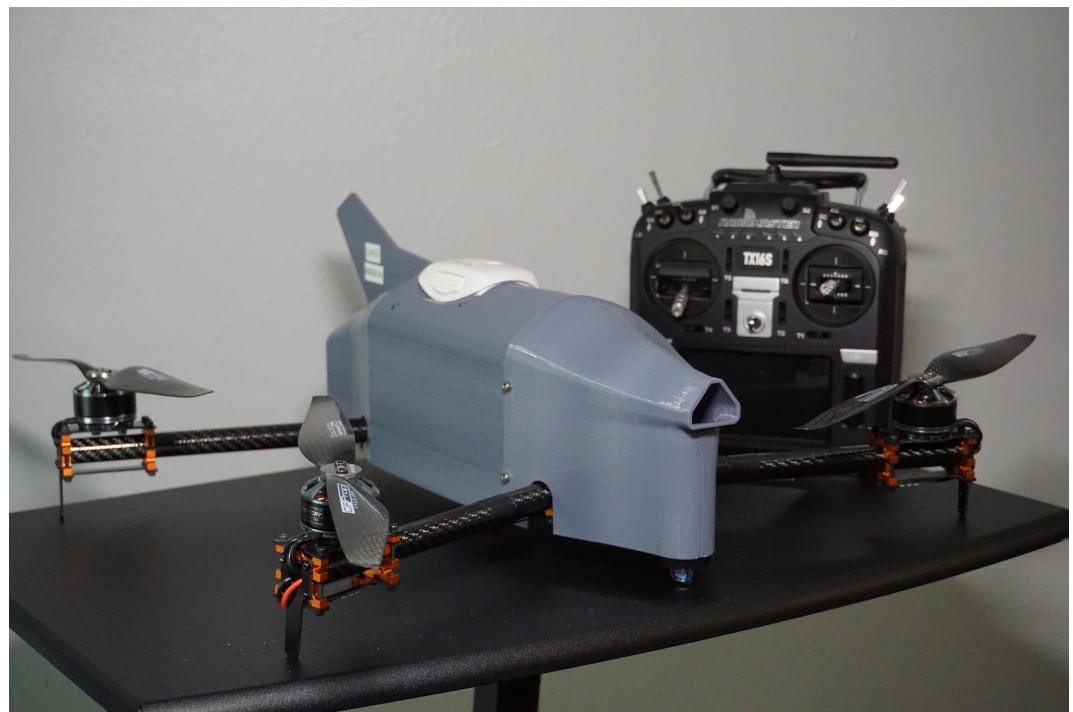

**Figure 2.** The CopterSonde UAS with radio controller

temperature sensors was selected based on findings in Greene et al. (2018). Within the intake scoop, there is a small fan to
aspirate the sensors, although it does not turn on until the CopterSonde reaches an elevation of 3 m above ground level (AGL) to
avoid ingesting dust into the scoop. In addition to the fan aspiration, the autopilot implements the wind vane mode explained in
Segales et al. (2020) to direct the scoop into the prevailing flow. As such, the air is not altered by the UAS before it passes over
the sensors. Additionally, the wind vane mode improves wind speed and direction estimation by increasing axis symmetry and
reducing vibrations. Wind speed and direction are determined by a linear algorithm estimator using aircraft attitude described
more in Section 4.2. Sensor accuracy response times, and further specifics on the system specifications can also be found in
Segales (2022).

## 3 Description of TRACER-UAS measurement locations, deployment strategies, and sampling

TRACER-UAS flights were completed at two locations south and southeast of Houston, TX, approximately 20 km from the
Gulf of Mexico, as seen in Fig. 3. The University of Houston Coastal Center (UHCC) site is a restored coastal prairie surrounded
by low-grade urban sprawl in La Marque, TX. This location lies 15 km due west of the Galveston Bay shoreline, and as a result,
frequently feels the effects of the bay breeze prior to the sea breeze. During TRACER, there was additional instrumentation
at UHCC, including a sonic anemometer and gas analyzer, ground-based remote sensors, and a sun photometer. The other

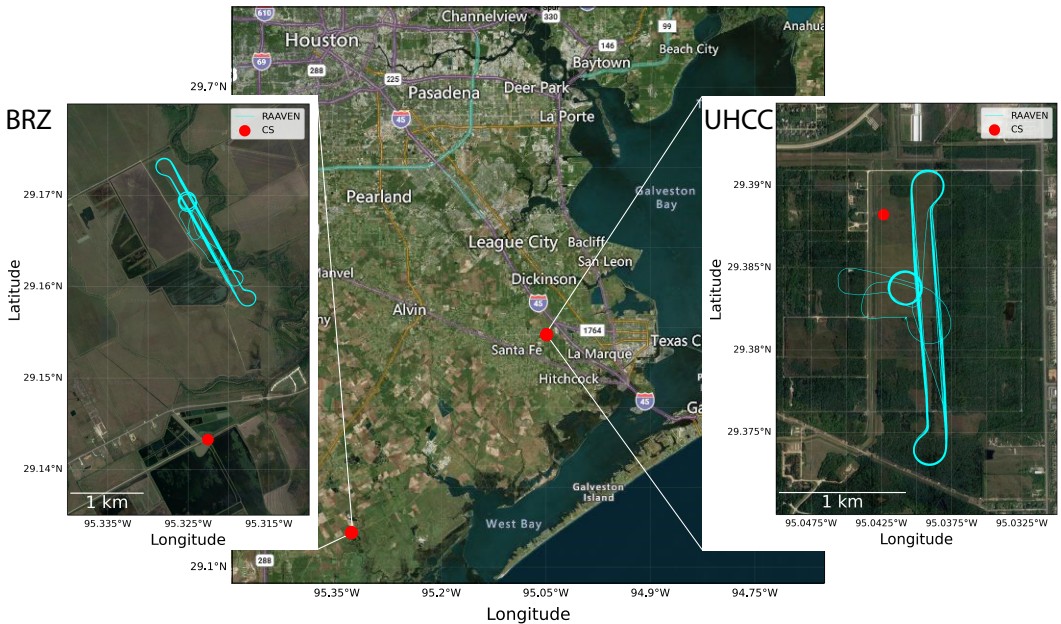

**Figure 3.** TRACER-UAS flight locations over a map of the Houston-Galveston area with zoomed-in inlays of the flight paths by the RAAVEN (blue line) and profiling site (red dot). The right and left maps are satellite imagery courtesy of ©Google Maps, 2022. The center map uses data from © OpenStreetMap contributors 2023. Distributed under the Open Data Commons Open Database License (ODbL) v1.0.

**June**

| | 1 | 2 | 11 | 12 | 13 | 14 | 15 | 16 | 17 | 18 | 19 | 20 | 21 | 22 | 23 | 24 | 25 | 26 | 27 | 28 | 29 | 30 | |
|---|---|---|---|---|---|---|---|---|---|---|---|---|---|---|---|---|---|---|---|---|---|---|---|
| CopterSonde | | | | | 17 | | 16 | 18 | 18 | | 10 | 19 | 18 | 18 | 18 | | | | | | | | UHCC |
| RAAVEN | | | | | | | | | | | 1 | 2 | 3 | 2 | 1 | 3 | 3 | 3 | 3 | 3 | 3 | | BRZ |

**July**

| | 1 | 2 | 11 | 12 | 13 | 14 | 15 | 16 | 17 | 18 | 19 | 20 | 21 | 22 | 23 | 24 | 25 | 26 | 27 | 28 | 29 | 30 | 31 |
|---|---|---|---|---|---|---|---|---|---|---|---|---|---|---|---|---|---|---|---|---|---|---|---|
| CopterSonde | | 16 | 14 | 17 | | | 18 | 20 | 16 | | | 20 | | | | | | | | | | | |
| RAAVEN | | | | | | | | | | | 3 | 2 | 2 | 3 | 2 | 2 | 4 | 3 | 3 | 2 | 3 | 3 | 2 |

**August**

| | 1 | 2 | 11 | 12 | 13 | 14 | 15 | 16 | 17 | 18 | 19 | 20 | 21 | 22 | 23 | 24 | 25 | 26 | 27 | 28 | 29 | 30 | 31 |
|---|---|---|---|---|---|---|---|---|---|---|---|---|---|---|---|---|---|---|---|---|---|---|---|
| CopterSonde | | | | | | | | | | | | | | 15 | 12 | | 16 | 16 | 15 | 17 | | 17 | 12 |
| RAAVEN | | | | | | | | | | | | | | | 3 | | 3 | 3 | 4 | 3 | 1 | 3 | 3 |

**September**

| | 1 | 2 | 11 | 12 | 13 | 14 | 15 | 16 | 17 | 18 | 19 | 20 | 21 | 22 | 23 | 24 | 25 | 26 | 27 | 28 | 29 | 30 |
|---|---|---|---|---|---|---|---|---|---|---|---|---|---|---|---|---|---|---|---|---|---|---|
| CopterSonde | | | | | | 19 | 16 | 15 | | 20 | 12 | 16 | 20 | 22 | | 13 | | | | | | |
| RAAVEN | 3 | 3 | | | | | | | | | 2 | 3 | 3 | 1 | 3 | 2 | 2 | 3 | 3 | 3 | 3 | 3 |

**Figure 4.** Data availability for each platform throughout TRACER-UAS in 2022, color-coded by the flight location with green indicating BRZ and yellow indicating UHCC. Numbers in each grid indicate the number of flights completed.

site near the Brazoria Wildlife Refuge (BRZ) is surrounded by wetlands and bayous southeast of Angleton, TX. At BRZ, the RAAVEN launch site is 2.7 km north of the CopterSonde for logistical reasons, while at UHCC, the flight tracks are much

closer (Fig. 3). Exact flight coordinates can be found within the data files.

A total of 4 IOPs, lasting two weeks in each month from June-September, were completed by both teams. Figure 4 outlines the data availability throughout the campaign. The CopterSonde team arrived one week before the RAAVEN team, such that there was one week of overlap to collect co-located observations each month. From June-August, the RAAVEN collected data only at the BRZ site. The CopterSonde collected data only at BRZ in July but used either site in June and August, depending

on the research objectives and weather conditions. In September, both teams only flew at UHCC due to landowner agreements. Table 1 documents the flight numbers for the campaign. In total, there are 13 days of co-located observations from both platforms.

Rotary-wing and fixed-wing UAS have distinct advantages that lead to different flight strategies. Figure 5 provides an overview of the altitudes and times of day (UTC) sampled by each of the two platforms. Flights were conducted during daylight hours in the altitude range spanning from the surface to 609 m AGL. These flights were supported by Certificates of Authorization (COAs) from the US Federal Aviation Administration (FAA). These distributions clearly illustrate that the two aircraft were operated under different sampling modes, as described below.

When equipped with the POPS sensor, the RAAVEN can fly up to 1.5 h, so the primary flight pattern combined helical profiles with long horizontal transects at multiple height levels to gather observations over a wider spatial region. During TRACER-UAS, the RAAVEN team would generally conduct three flights daily, with each flight starting with a profile up to 600 m AGL and would then proceed to complete a series of stepped-level legs, where the aircraft would maintain an altitude for approximately 9 minutes per leg. The altitudes sampled by these level legs during TRACER were nominally 600, 400, 250, 150, 100, 50, and 20 m AGL, though sometimes adjustments were needed due to weather conditions or air traffic conflicts. After the completion of these level legs, the aircraft would conduct another profile or two to 600 m AGL before landing to end the flight.

The CopterSonde has a shorter battery life, but batteries can be quickly replaced to conduct vertical profiles with high temporal resolution. Figure 6 shows the typical altitude flight pattern for the RAAVEN with CopterSonde flight cadence on a co-located observation day. Each flight up to 609 m takes about 6 min to complete, which is how on 22-23 September, there were 4 flights completed at a 7-min cadence during a late-onset sea breeze. Typically, the flight cadence was 30 min until there was evidence of a sea breeze moving onshore or interesting features in the temperature profile that would motivate increasing the flight cadence to 15 min. Most days had at least 1 h of flights at a 15-min cadence, which was decided in real-time using satellite and CopterSonde data. Given the flexibility of the flight strategy, the start and end time were chosen two days in advance to meet certain objectives. The FAA only allowed 8 h of flight time per pilot per day, so operations usually started between 0800-1000 LST and ended at 1600-1800 LST. Team members used numerical weather models to estimate the sea breeze timing and convection initiation to decide operation hours. Throughout the dataset, there are some breaks in the flight

| Aircraft | CU RAAVEN | OU CopterSonde |
|---|---|---|
| Flight days at UHCC | 12 | 19 |
| Flight days at BRZ | 35 | 14 |
| Total # of flights | 131 | 549 |
| # of Profiles | 251 | 547 |
| Flight hours | 187 | 56 |

**Table 1.** Flight statistics from the CU RAAVEN and OU CopterSonde across the entire TRACER-UAS campaign

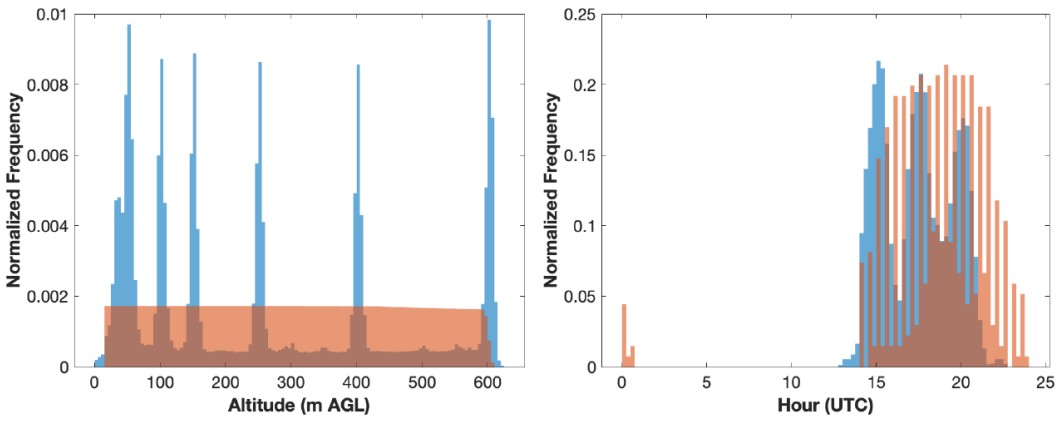

**Figure 5.** Histograms illustrating the altitudes (left) and hours of day (right) sampled by the RAAVEN (blue) and CopterSonde (red) during the TRACER-UAS campaign.

pattern due to lightning or rain delays, technical errors, or airspace deconfliction. Low clouds were an occasional problem that limited the flight ceiling, but 75% of CopterSonde flights reached the 609 m flight ceiling.

## 4 Data processing and quality control

### 4.1 University of Colorado RAAVEN

Data collected by the RAAVEN's sensors during TRACER-UAS were logged at a variety of different logging rates. As with previous deployments, the finewire system was logged at 250 Hz, the fastest rate of all of the sensors. The BST MHP was logged at 100 Hz, the VectorNav VN-300 at 50 Hz, the Melexis IR sensors and variables related to finewire status at 20 Hz, data collected from the PixHawk autopilot and Vaisala RSS421 sensors at 5 Hz, and data from the POPS aerosol spectrometer, a new addition for this campaign, at 1 Hz. All logging events carried out by the FlexLogger include a sample time from the

logger CPU clock, allowing for post-collection time alignment between the different sensors. A detailed description of the time alignment process is included in Cleary et al. (2022).

For the B1-level data files, the re-sampled (in time) data include several derived and measured quantities, which are provided at 10 Hz. These data include information on aircraft position, including information on latitude, longitude, and altitude, as measured by the VN-300. Aircraft altitude is corrected using a combination of various inputs from onboard GPS and pressure

altimeters, as neither of these altitude estimates can be used reliably as a definite flight altitude. Information on derivation of the aircraft altitude is also provided in Cleary et al. (2022). As with previous campaigns, a flight_flag binary variable is developed by combining information on aircraft airspeed and altitude, as provided by the autopilot system. Times where the aircraft airspeed exceeds 10 m s$^{-1}$ and the aircraft altitude is greater than 5 m AGL are flagged as periods where the RAAVEN is flying (flight_flag = 1). The time point 4 s (200 samples) before the first point where flight_flag is set to 1 is recorded as the

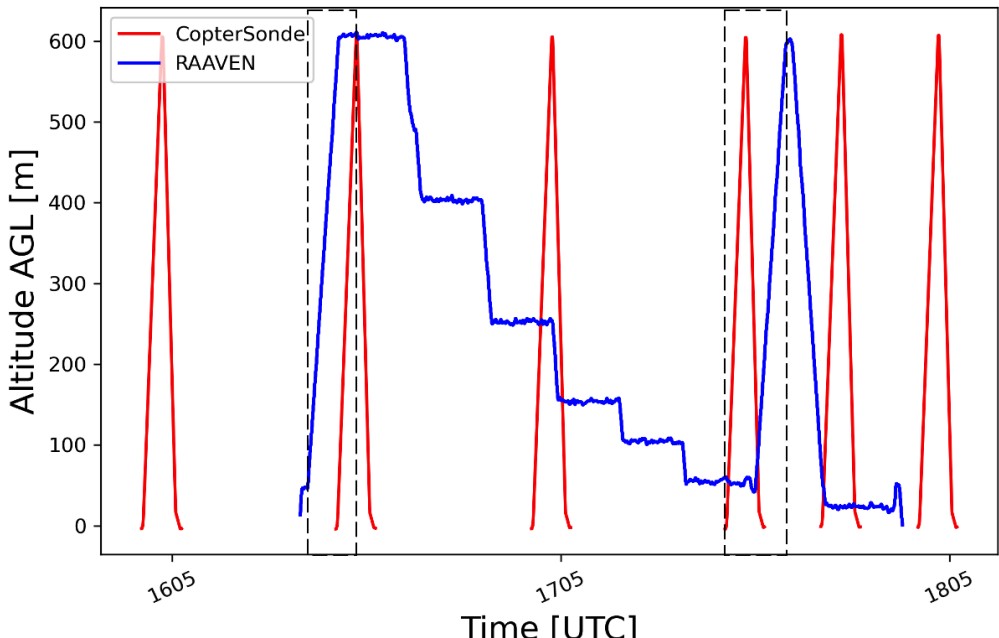

**Figure 6.** Typical flight altitude patterns from RAAVEN (blue) and CopterSonde (red) during a single RAAVEN flight. Dashed boxes highlight examples of vertical profiles pulled from each platform for the data comparison.

take-off point, while the time point 4 s (200 samples) after the last flying point in the record is designated as the index where the aircraft has landed.

Deriving wind information from fixed-wing research aircraft systems is a complex undertaking (see van den Kroonenberg et al., 2008). Doing so requires the combination of information from different sensors, including measured airspeed, airflow angle over the aircraft, and aircraft motion relative to the Earth system. For the RAAVEN platform, any biases in true airspeed (TAS) can impart significant errors in the calculation of wind velocity, while time-lag between the reported GPS velocities and in-situ measured aircraft attitude, and any angular offsets between the INS and MHP tend to have smaller impacts. In this study, these potential sources of error are corrected for by implementing an optimization technique. In this technique, small adjustments are made to individual parameters including airspeed, angle of attack, sideslip angle, and temporal logging offset to generate a wind solution. Then these individual wind solutions are evaluated and the one with the smallest overall sinusoidal variability over individual orbits or racetracks is selected as the correct combination for deriving wind parameters (see Cleary et al. (2022) for full details).

To improve the usability of the parameters measured during the TRACER campaign, the datafiles developed as part of the RAAVEN dataset have been assigned data quality flags. These flags are determined through a variety of means, as described here. The flag associated with the RSS421-derived temperature is set to zero for time periods that are deemed to consist of good data and set to 1 for times when there are potential data quality issues, as identified by: (a) the absolute value of the difference

between the temperature from either individual sensor being greater than 0.5 ℃, (b) the absolute value of the difference between the RSS421 temperature and the temperature from the EE-03 sensor on the MHP exceeding 5 ℃, (c) the internal error flag of either RSS421 sensor being active, or (d) the aircraft not being in the flying state identified using the flight_flag parameter. For the RH measurement from the RSS421, similar criteria are implemented, except limits are set to be 5 % between RSS421 sensors and 15 % between the output RH value and the MHP-provided RH value. This second value is as large as it is because the RH values from the MHP-mounted sensor are impacted by exposure of that sensor to sunlight, and the associated impact on sensor temperature. Because these temperature swings are not corrected for, this MHP-mounted sensor can produce large fluctuations in the RH values. As a result, this MHP-based RH measurement is only meant to provide a reality check to ensure that the RSS421 sensors are reporting accurate values. The most important comparison is between the two RSS421 sensors, which should agree much more closely, as they are the same sensor type and are mounted within close proximity of one another.

In addition to the RSS421 flags, there is also a data quality flag implemented for the coldwire temperature sensor. This data quality flag is activated when the difference between the coldwire-derived temperature value and either RSS421 temperature exceeds 0.6 ℃, when the absolute value of the difference between the coldwire-derived temperature and that from the MHP-mounted sensor exceeds 2 ℃, when coldwire voltages are observed to fall outside of the 0–4 V analog range, or when flight_flag is zero. There is also a pressure quality control flag for the pressure measurements from the VN-300. This flag is activated if the absolute value of the difference between the VN-300 static pressure and that measured by either RSS421 sensor exceeds 4 hPa. The RSS421 pressure measurements are not used as the primary pressure measurement because comparisons with radiosonde and tower data indicate that they are likely biased low due to the airflow passing over their location on the aircraft.

The RAAVEN dataset also includes a three-stage data quality flag for wind estimates from this platform. This flag is set to 0 for times where wind data are deemed to be good, 1 for time periods where data are potentially suspect, and 2 where data are known to be of poor quality. Data are labeled to be bad if any of the following are met:

- Measured angle of attack or sideslip exceeds 20 degrees. Times where angle of attack or sideslip are between 10–20 degrees are flagged as "suspect". This is because the multihole probe is calibrated to +/- 15 degrees angle of attack or angle of sideslip.

- Measured true airspeed (TAS) is less than 10 m s$^{-1}$.

- There is noted blockage of MHP ports, as indicated by differential pressure values reported by the MHP falling below -100 Pa.

- The 40 second moving average of a 20 second moving window variance of the MHP-derived TAS is above 5 m$^2$s$^{-2}$.

- The flight_flag is zero.

Data users should be aware that extending multihole probe calibration coefficients beyond angles tested in the wind tunnel can result in highly non-linear errors in estimation of angle of attack and sideslip angle. Such errors significantly impact wind estimation. As such users are advised to only use wind data where the wind_flag variable is equal to zero. These values are well within the calibrated range of the multihole probe.

Because the variables associated with the POPS dataset might be challenging for some data users to interpret and understand, we here provide an overview of the variables associated with this sensor. To calculate flow-corrected particle concentrations from the POPS data, the number of particles counted in each bin is divided by measured flow rate in the POPS sensor multiplied by a ratio of the ambient temperature to the sensor temperature:

$$N_a^x = \frac{X_a^x}{F_{POPS} * \frac{T_a}{T_{POPS}}}$$

where $N_a^x$ is the flow-corrected particle concentration for bin x, $X_a^x$ is the POPS particle count for bin x, $F_{POPS}$ is the flow rate measured for POPS, $T_a$ is the temperature of the ambient air, and $T_{POPS}$ is the laminar flow element temperature (POPS_Temperature in the dataset). This flow-corrected variable (POPS_partconc for the total aerosol concentration, or POPS_binX_partconc for bin-resolved particle concentrations) is the recommended variable for determining the aerosol number concentration from RAAVEN. Additionally, the NetCDF files provide information on the standard deviation of POPS measurements over one second (POPS_STD), the pressure inside POPS (POPS_Pressure), the instrument flow rate (POPS_Flow), the laser diode monitor and temperature (POPS_LDM and POPS_LDtemp), the combined total particle count (POPS_HistSUm), and a flag for the implementation of a manual binning process (POPS_useman). To allow users to plot size distributions, there is a variable that describes the edges of the 24 bins used by the POPS (POPS_Bin_Edges). There is also a flag included in the TRACER-UAS datastream for the POPS aerosol spectrometer. This flag is based on different values for the ones, tens, hundreds, and thousands places. The ones digit is set to 0 if data are ok, and 1 if either the aircraft is not in flight or the inlet filter is suspected to be in place. The tens digit reacts to the temperature of the sensor. The POPS is designed to function optimally at temperatures below 45 ºC, and increasing temperatures impact the measurement uncertainty. Therefore, the flag is set to 0 if the temperature is less than 45 ºC, 1 if the temperature is between 45-48 ºC (uncertainty < 3 %), 2 if the temperature is between 48-50 ºC (uncertainty < 7 %), and 3 if the temperature is greater than 50 ºC (do not use, uncertainty too high). The hundreds place is set based on the observed standard deviation of the measurement, with lower standard deviations being assigned 0 (good data), and standard deviations above 14 being assigned 1. Under higher standard deviations, the user is advised to use the first two bins with caution, as uncertainty at the smaller size ranges can be up to 35 % in this case. Finally, the thousands place is assigned based on the flow rate of the airstream being sampled. This flag is set to 0 for good data, and set to 1 when the flow rate is lower than 2 cm$^3$ s$^{-1}$, as the lower flow rate increases uncertainty in the measured quantities.

Finally, there are two additional flags included in the RAAVEN data files to allow data users to easily understand the aircraft's flight state and support selective sampling during specific flight regimes. These flags include the "Flight_Flag" introduced earlier in the manuscript, as well as a second "Flight_State" flag, which is a three-symbol binary variable. The Flight_State flag offers insight into whether RAAVEN is flying straight (0 in the ones place) or is turning (1 in the ones place), whether RAAVEN is descending (0 in tens place), level (1 in tens place), or ascending (2 in tens place), and whether RAAVEN is in flight (1 in hundreds place) or not (0 in hundreds place). For example, if a data user wanted to analyze straight, level flight legs, they would search for data with Flight_State equal to 110.

The accuracy of the RAAVEN observations has been evaluated in previous studies. This includes a comparison of RAAVEN data with measurements collected by radiosondes launched from the Barbados Cloud Observatory (de Boer et al., 2022) and comparisons supported by radiosonde and tower data collected by the US DOE ARM SGP facility (de Boer et al., 2023).

## 4.2 University of Oklahoma CopterSonde

Raw data from the CopterSonde are stored on an SD card as binary files and then converted to a0-level NetCDF. Subsequently, data go through a process of averaging, filtering, and objective quality analysis to optimize the quality of observations. Since data from the Pixhawk are logged at a faster rate of 20 Hz than the temperature and humidity sampling rate of 10 Hz, the position and rotation data gathered by the Pixhawk are downsampled to 10 Hz to ensure a standard timeline of observations. After achieving a common time coordinate, offsets determined in the Oklahoma Climatological Survey calibration chamber are applied to each sensor. Every CopterSonde is calibrated prior to deployment, so each one has unique offsets due to minor differences in sensors. To eliminate spurious, high-frequency signals in data, the attitude data (roll, pitch, and yaw), temperature, and relative humidity data have a low-pass finite impulse response (FIR) filter applied, described in Greene et al. (2022). Sets of three identical temperature and humidity sensors were used to ensure agreement between observations. Acceptable thresholds for sensor bias and standard deviation were experimentally determined during sensor calibration and characterization studies; if an individual sensor exceeds those thresholds, the sensor's observations are removed. Then, all remaining thermistors and humidity capacitors are averaged, and the data are binned to 5 m. This combination of sampling rate and vertical resolution ensures at least 16 observations per sensor in each bin.

Wind direction is estimated during the flight by altering the yaw angle to minimize the roll to optimize stability and promote flow into the sensor scoop. A wind speed estimate comes from the pitch angle, and in cases of high wind, the UAS will automatically return home to avoid battery fatigue or failure. In post-processing, a more accurate horizontal wind vector is derived using a more robust linear model on the roll, pitch, and yaw while accounting for the aircraft geometry. In cases of very low wind speeds, the autopilot struggles to calculate the true wind direction, and for wind speeds less than $2$ m s$^{-1}$ the wind direction values are considered questionable. In data comparison calculations, the wind speed and direction were removed when the wind speed was less than $2$ m s$^{-1}$ (Table 2).

## 5 System Intercomparison

The operational dates for each platform were intentionally staggered by a week to expand the amount of data collected by the two platforms over the course of the four-month campaign. Nevertheless, on 13 days, both teams were co-located and conducting simultaneous flight operations offering a limited amount of data for platform intercomparison. Such intercomparison was viewed to be important to ensure that there was no significant sensor drift over the four-month field campaign window, and to ensure that data from the two platforms was able to be used confidently together to develop statistics. This intercomparison included the calculation of mean vertical profiles from both platforms during time periods when both aircraft captured profiles within 15 minutes of one another. Such tight time alignment was required to minimize the impacts of a rapidly evolving atmo-

| Base Variables | Correlation | Mean Difference (CS-RAAVEN) | $\sigma$ | Uncertainty or Estimated Bias (CS/RAAVEN) |
|---|---|---|---|---|
| Temperature [K] | 0.985 | -0.287 | 0.327 | 0.3/0.2* |
| Relative Humidity [%] | 0.945 | -3.763 | 3.934 | 3/3* |
| Wind Direction [º] | 0.967 | 0.067 | 13.256 | 1.15/-3.76[†] |
| Wind Speed [m s$^{-1}$] | 0.829 | -0.409 | 1.227 | -0.70/0.58[†] |

**Table 2.** Data comparison statistics from 4425 data points of co-located vertical profiles from the RAAVEN and CopterSonde (CS). Values with an asterisk are sensor uncertainties provided by the manufacturer. Values with an [†] are mean biases based on comparisons to radiosondes reproduced from de Boer et al. (2023).

spheric boundary layer. A total of 44 profiles from each platform were matched and used in the data comparison. Moreover, the RAAVEN data were interpolated to a 5-m vertical grid to match the vertical resolution of the CopterSonde, and both were set to an equal profile depth. Table 2 provides the statistical comparison of direct observations from each platform. Temperature, relative humidity, and wind direction are all in strong agreement with Pearson correlations above 0.9. The wind speed correlation is lower due in part that the CopterSonde wind speed calculation is less accurate at very low wind speeds, and the relatively high amount of variability in wind speed and direction within the eddy-driven structure of a convective boundary layer with low mean wind speeds. Because near-zero wind speeds are challenging to compare, the wind data were filtered if the wind speed was less than 2 m s$^{-1}$. As an extra assurance of data quality, the CopterSonde wind speed and direction were compared against a Doppler lidar located within 6 m of the profile site at the UH Coastal Center (comparison not shown). The lidar completed 60º plan-position indicator scans every 15 min in order to calculate velocity azimuthal displays of wind speed and direction. The correlation between the lidar winds and the CopterSonde is 0.866, which is slightly better than the correlation between the two UAS platforms, but the mean difference shows a 0.326 m s$^{-1}$ underestimation of wind speed by the CopterSonde.

Figure 7 shows all of the profiles collected within 15 minutes of one another while both platforms were operating at the same flight location (BRZ or UHCC). It should be noted that in these instances there was approximately 2500 m separating the aircraft at BRZ and around 500 m separating the aircraft at UHCC, so some spatial sampling differences can be expected between the two aircraft. As a result of these differences, we do not expect a perfect correlation between the two platforms. This figure is primarily meant to demonstrate that quantities measured by the two platforms are generally comparable over the course of the campaign, and that there isn't a significant drift in either platform's sensors over the four months of the IOP. In most cases, data shown here are in good agreement, with enhanced scatter in the wind speeds and humidity values as might be expected in a convective boundary layer. Additionally, CopterSonde provided temperatures are shown to be consistently lower than those from the RAAVEN. This result is consistent with the more detailed evaluation provided in (de Boer et al., 2023), where CopterSonde data are shown to be biased by -0.12 K and RAAVEN data are biased by 0.31 K, relative to radiosondes. The data shown in Figure 7 illustrate the impact of heterogeneity in the ABL over short distances, together with small differences

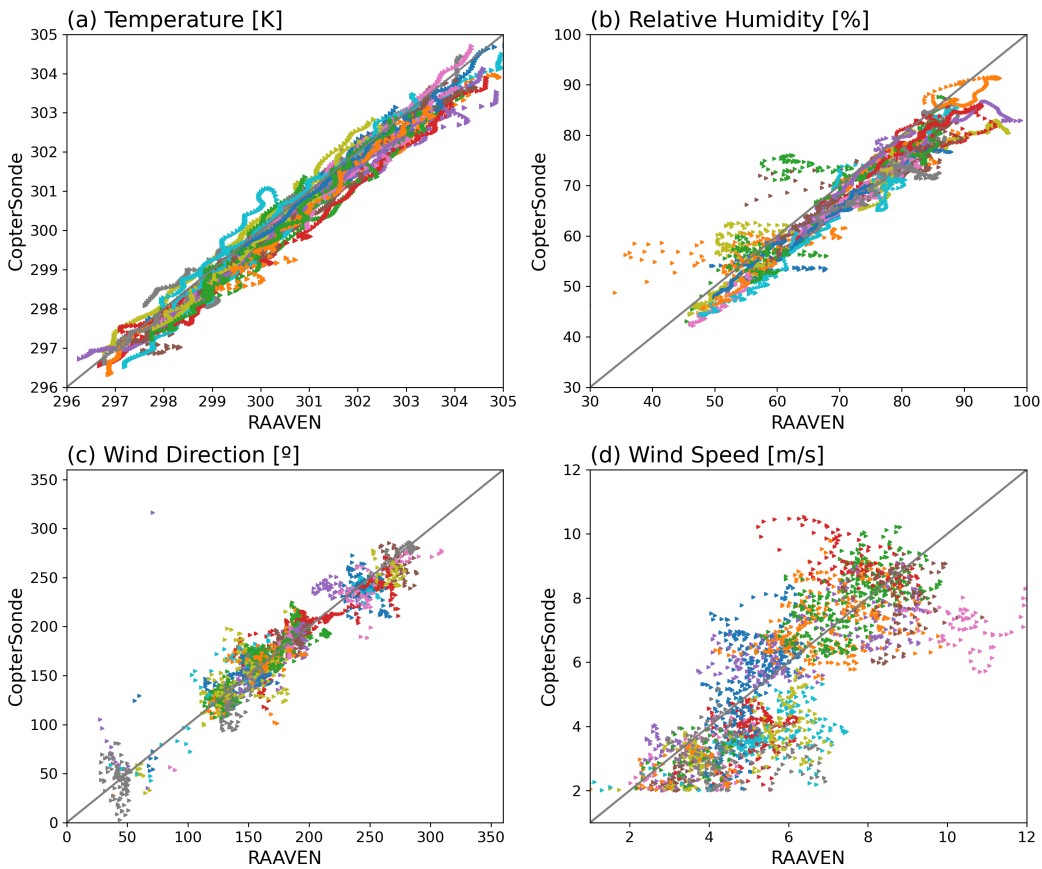

**Figure 7.** Scatter points of all vertical profiles comparing both data platform observations of (a) temperature (K), (b) relative humidity (%), (c) wind direction (degrees) and (d) wind speed (m s$^{-1}$). Each color represents data points from a separate profile, and the gray line indicates a 1:1 slope.

in sensor performance and system biases. The observations from each platform provide unique and complementary data to be used to capture the micrometeorology of the coastal region.

It is important to note that this intercomparison is not meant to be a detailed evaluation of the measurement quality provided by both platforms. Such an evaluation was previously conducted in the pre-campaign preparations for the TRACER project, and the results of that evaluation can be reviewed in de Boer et al. (2023). Overall, differences in observations in the TRACER-UAS dataset fall in line with findings from that manuscript. While the data are not perfectly correlated due to inherent spatial heterogeneity in the ABL and differences in sensors, the data are in strong agreement with each other and ground-based observations, offering confidence that the datasets can be used together to develop statistical analyses of atmospheric phenomena as part of this deployment.

## 6 Overview of Sampled Conditions

The TRACER-UAS observing periods occurred under drought conditions, pluvial events, and seasonal sea breeze conditions, resulting in a variety of conditions sampled. Throughout June and July, the region was under severe to extreme drought conditions (USDM, Svoboda et al. (2002)), but enhanced sea breeze convection in July and synoptically-forced rainfall in August led to a decline in drought severity later in the summer. Figure 8 shows the range and frequency of observations collected by the RAAVEN and CopterSonde. Conditions were overall warm and relatively humid. A typical daily pattern at either of the sampling locations started with (relatively) cooler and very humid conditions at the surface, under the development of an early morning boundary layer that the aircraft would be able to sample through into a residual stable layer aloft. This boundary layer would transition quickly as a result of strong solar warming of the surface, increasing temperatures, and decreasing relative humidity at the surface. Small fair-weather cumulus clouds would form and deepen throughout the morning, with the SBF passing through around mid-day. This frontal passage helped to invigorate convection along its boundary, after which the onshore flow would be established. While there would typically be an increase in winds and a shift in wind direction with this transition, there was not a significant temperature signature. However, during the sea breeze was frequently devoid of cloud cover. Over the course of the campaign, winds were generally light, and spanned the full 360º range of possible wind directions. However, there is a clear peak in the wind directions measured around 150-180 degrees (southeast), signifying the wind direction under sea breeze conditions. Also notable is the fact that the RAAVEN conducted flights in late September after CopterSonde flights had been completed that featured cooler, drier, and more northerly wind conditions. Finally, there was a significant range of different aerosol regimes samples, including very clear conditions, as well as polluted conditions that were associated with a variety of different wind conditions. These polluted conditions were associated with both local industrial activities related to regional oil and gas production, emissions from the city of Houston to the north, and emissions from local wildfires. One flight, the RAAVEN was regularly flying in and out of a wildfire smoke plume from a fire that had been established approximately 4.5 miles from the flight operations site.

## 7 Data Availability

TRACER-UAS CopterSonde and RAAVEN data are available through the ARM data center (https://doi.org/10.5439/1969004 (Lappin, 2023) and https://doi.org/10.5439/1985470 (de Boer, 2023)). The ARM data center requires a free ARM user account to access either dataset (https://adc.arm.gov/armuserreg/#/new). Reviewers can access a public directory at https://adc.arm.gov/essd/tracer-uas/. All files come in NetCDF format with the naming convention of [location]_[platform]_tracer_[data level]_YYYYMMDD.HHMMSS.nc. The CopterSonde data files have a prefix of ARM0735. The two location options are uhc (University of Houston Coastal Center) and brz (Brazoria Wildlife Refuge). The two platform options are coptersonde or CU-RAAVEN. CopterSonde data offer two file levels, a0 and c1, while the RAAVEN data are b1 level only. Tables A and A2 list all processed variables included in the RAAVEN and CopterSonde files. CopterSonde a0 level files include all raw data from each sensor and the Pixhawk autopilot, including pitch, roll, and yaw. Please contact the author for more information on the a0 variables. This work is licensed under the Creative Commons Attribution 4.0 International License.

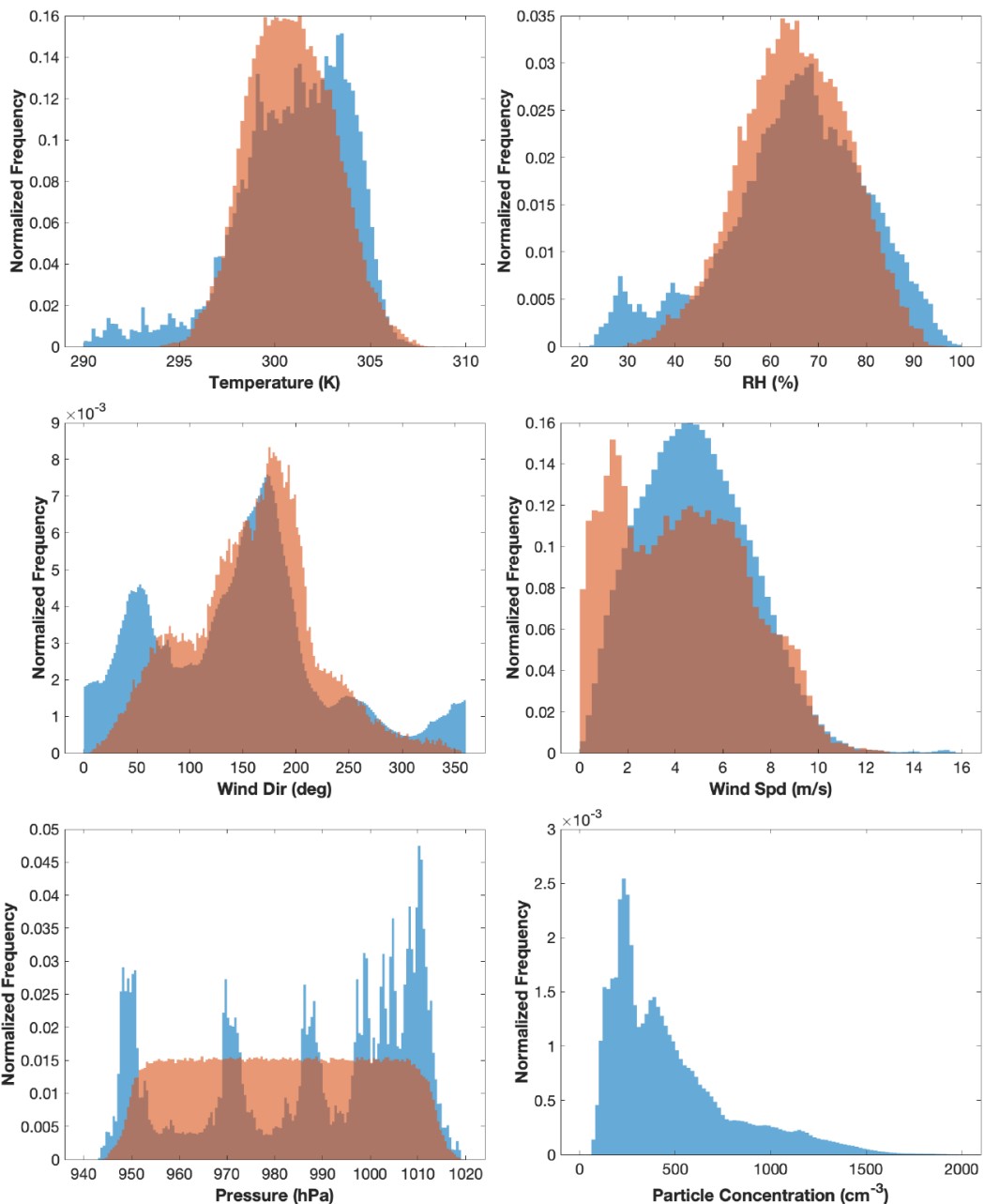

**Figure 8.** Histograms illustrating the range of conditions sampled by the RAAVEN (blue) and CopterSonde (red) during the TRACER-UAS campaign. Included are histograms of temperature, relative humidity, wind speed, wind direction, pressure, and particle concentration (RAAVEN only).

# 8  Summary

The TRACER-UAS campaign occurred from June-September south of Houston near the Gulf of Mexico coastline. Two UAS platforms were employed to sample the ABL at high spatial and temporal frequencies under conditions including the SBC, through storm evolution, and under quiescent ABL conditions. The RAAVEN and CopterSonde collected over 200 h of flight data across 61 days up to 609 m. Teams were frequently co-located within a kilometer or two of each other to get a four-dimensional view of the ABL using both platforms. Each platform collected thermodynamic and kinematic observations, with the RAAVEN additionally gathering aerosol size distribution and brightness temperatures. These observations complement the TRACER campaign by delivering four-dimensional, lower-atmospheric observations of local circulations and their interactions with convection, as well as quiescent periods. All data were processed and quality analyzed to ensure high validity and precision. Observations from each platform have shown to agree well with each other (Table 2), allowing the complementary use of datasets to understand ABL characteristics and evolution with respect to the convective cloud lifecycle, the SBC, and pre and post-storm processes. The utility of these observations also extends to contextualizing air quality and pollutant transport and their interactions with clouds and precipitation. TRACER-UAS observations offer a unique component to the broader TRACER campaign through a dense dataset in the commonly undersampled ABL.

## Appendix A

Table A1: List of all variables included in RAAVEN B1 files with units and respective sensors. Please note that additional information on these variables is available in the metadata that is included in the NetCDF files that comprise the dataset.

| Variable name | Units | Sensor |
| --- | --- | --- |
| time | seconds since 2020-01-01 00:00:00:00 | VectorNav |
| base_time | seconds since 2020-01-01 00:00:00 UTC | VectorNav |
| time_offset | seconds since base_time | VectoNav |
| time_10hz | seconds since midnight | Interpolated |
| Flight_Flag | unitless | multi-sensor |
| Flight_State | unitless | multi-sensor |
| alt | meter | Pixhawk and VectorNav |
| lat | degrees | Pixhawk |
| lon | degrees | Pixhawk |
| yaw | degrees | Pixhawk |
| pitch | degrees | Pixhawk |
| roll | degrees | Pixhawk |
| air_temperature | Kelvin | Vaisala RSS-421 |
| air_temperature_flag | unitless | multisensor |
| air_temperature_fast | Kelvin | Vaisala RSS-421 and cold wire |
| air_temperature_fast_flag | unitless | multisensor |
| relative_humidity | % | Vaisala RSS-421 |
| relative_humidity_flag | unitless | multisensor |
| air_pressure | hPa | PixHawk |
| air_pressure_flag | unitless | multisensor |
| alpha | degrees | Multihole Probe |
| beta | degrees | Multihole Probe |
| eastward_wind | m s$^{-1}$ | multisensor |
| nortwward_wind | m s$^{-1}$ | multisensor |
| vertical_wind | m s$^{-1}$ | multisensor |
| wind_speed | m s$^{-1}$ | multisensor |
| wind_direction | degrees | multisensor |
| TAS | m s$^{-1}$ | Multihole Probe |

| | | |
|---|---|---|
| VE | m s$^{-1}$ | VectorNav |
| VN | m s$^{-1}$ | VectorNav |
| VD | m s$^{-1}$ | VectorNav |
| wind_flag | unitless | multisensor |
| brightness_temperature_sky | Kelvin | Melexis |
| brightness_temperature_surface | Kelvin | Melexis |
| POPS_STD | unitless | POPS |
| POPS_Pressure | hPa | POPS |
| POPS_Temperature | Celcius | POPS |
| POPS_Flow | cm$^3$ s$^{-1}$ | POPS |
| POPS_LDM | unitless | POPS |
| POPS_LDtemp | Celcius | POPS |
| POPS_binXX | s$^{-1}$ | POPS |
| POPS_HistSum | s$^{-1}$ | POPS |
| POPS_useman | unitless | POPS |
| POPS_partconc | cm$^{-3}$ | POPS |
| POPS_binXX_partconc | cm$^{-3}$ | POPS |
| POPS_Bin_Edges | nm | POPS |
| POPS_flag | unitless | multisensor |

**Table A2.** List of all variables included in CopterSonde c1 files with units and respective sensors

| Variable name | Units | Sensor |
| --- | --- | --- |
| time | microseconds since 2010-01-01 00:00:00:00 | Pixhawk |
| base_time | seconds since 1970-01-01 00:00:00 UTC | Pixhawk |
| time_offset | seconds since base_time | Pixhawk |
| alt | meter | Pixhawk |
| pres | pascal | MS5611 |
| lat | degree | Pixhawk |
| lon | degree | Pixhawk |
| tdry | kelvin | iMet-XF bead thermistor |
| mr | $kg\ kg^{-1}$ | Derived from temperature, pressure, and relative humidity sensors |
| theta | kelvin | Derived from temperature and pressure sensors |
| Td | degree Celsius | Derived from temperature and relative humidity sensors |
| q | $g\ kg^{-1}$ | Derived from temperature, pressure, and relative humidity sensors |
| rh | % | HYT-271 capacitive humidity sensor |
| dir | degree | Pixhawk |
| wspd | $m\ s^{-1}$ | Pixhawk |
| wind_u | $m\ s^{-1}$ | Pixhawk |
| wind_v | $m\ s^{-1}$ | Pixhawk |

*Author contributions.* All co-authors contributed to the generation of the TRACER-UAS dataset. GB and EPL coordinated data collection
and airspace authorizations. GB, JH, RC, MR, and JB supported data collection with the RAAVEN. FL, MS, EPL, IM, LO, KB, BP, AJ, AS,
and ES aided in data collection with the CopterSonde. RAAVEN data processing and quality control were completed by GB, RC, BB, and
EA. CopterSonde data processing and quality control were completed by FL. CopterSonde equipment was prepared and maintained by AS.
FL and GB collaborated on the development of the manuscript with editing support from all other co-authors.

*Competing interests.* The authors declare that they have no conflict of interest

*Acknowledgements.* This work was supported by the US Department of Energy (DE-SC0021381). Additional support was provided by the
NOAA Physical Sciences Laboratory. We would like to acknowledge the support of the University of Houston and from private landowners
who provided access to property to conduct flight operations.

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
