# Peer review of "Data collected using small uncrewed aircraft system during the TRacking Aerosol Convection Interactions ExpeRiment (TRACER)"

_Earth System Science Data, 2023_

## Author Comment (AC1)

**Authors' Response to Reviews of**

**Data collected using small uncrewed aircraft system during the TRacking Aerosol Convection Interactions ExpeRiment (TRACER)**

Francesca Lappin, Gijs de Boer, Petra Klein, Jonathan Hamilton, Michelle Spencer, Radiance Calmer, Antonio R. Segales, Michael Rhodes, Tyler M. Bell, Justin Buchli, Kelsey Britt, Elizabeth Asher, Isaac Medina, Brian Butterworth, Leia Otterstatter, Madison Ritsch, Bryony Puxley, Angelina Miller, Arianna Jordan, Ceu Gomez-Faulk, Elizabeth Smith, Steven Borenstein, Troy Thornberry, Brian Argrow, and Elizabeth Pillar-Little
*Earth System Science Data,* 2023-371
* * *
**RC:** *Reviewers' Comment*,     AR: Authors' Response,     ☐ Manuscript Text

**1.   Reviewer 1**

**1.1.   General Comments**

**RC:** *This paper provides 200 flight hours of data during the TRACE-UAS to support the project goal – further understanding the role that regional circulations and aerosol loading play in the convective cloud life cycle across the greater Houston, Texas area. The authors presented a very useful payload for the atmospheric study and flight conditions. The meteorological data is of high quality. However, the aerosol data are very limited, and the data quality is still unknown. The paper heavily focused on the met data discussion, which is great but missing the connection to support half of the project goal.*

 AR:   We are glad to hear the reviewer found the observations to be both reliable and useful. The authors have increased the description of the aerosol data collected throughout the manuscript. The connection of the aerosol data to the main project was expanded in the introduction. The description of the aerosol data was expanded in the RAAVEN data processing section. Regarding the quality of the aerosol measurements, there are several publications that highlight the performance of POPS. For this project, the POPS was calibrated prior to the field deployment, and the data quality flag for that instrument provides additional information on the expected accuracy, given several potential instrument issues in this operational environment. We have added text to the manuscript to offer more insight into the quality of these measurements (see paragraphs starting on line 103, and line 245.

**1.2.   Specific Comments**

**RC:** *Introduction: this session highlighted the observational gap in aerosol and gas phase measurements but didn't mention the importance of the combined datasets. I also recommend including why it is essential to understand thermodynamic and kinematic data and their linkages to the aerosol properties/distribution in the region.*

 AR:   Lines 63-67 were added to expand on the interconnections of aerosol and trace gases as a function of the thermodynamic and kinematic evolution within the ABL. Additional references were included to provide more context for the motivations of filling these data gaps.

**RC:** *P5, line 99-100, what is the aerosol collection efficiency of the platform? Does the flight orientation affect*

*the aerosol collection? How do you validate the aerosol data accuracy with the platform?*

AR:   Thank you for raising this important point. We have added information on the calculated sampling efficiency and other items related to uncertainty. Specifically, this text reads: "The POPS provides information on aerosol size distribution, giving size-resolved concentrations across 24 size bins, as well as information on a variety of instrument monitoring systems. POPS data are available for all but four of the completed flights, during which sensor overheating resulted in no usable data. This overheating issue was resolved by adding additional ventilation to the payload bay carrying the POPS sensor once it was discovered. POPS data quality has been evaluated in previous studies Mei et al., 2020. The inlet for the POPS instrument was located on the RAAVEN fuselage and consisted of a 2 mm ID (3 mm OD) brass tube mounted in an isoaxial configuration. The POPS was operated at a sample flow rate of 3 cm$^3$ s$^{-1}$, resulting in sub-isokinetic sampling at RAAVEN airspeeds and leading to size dependent oversampling of particles. The sample line between the inlet and the instrument was constructed of brass and conductive silicone tubing. Overall particle sampling efficiency accounting for aspiration and transmission Baron and Willeke, 2001 ranged from 1.1 to 2.1 over the POPS measurement size range. Data quality can additionally be impacted by a variety of different things, including sensor temperature and flow rate. These issues are diagnosed in post-processing, and the POPS data quality flag (see below) helps provide users with additional information on sensor data quality."

Additionally, there is information in the processing section on other variables used to detect potential data quality issues. Specifically, this includes sensor temperature, sensor flow rate, and measured standard deviation. These thresholds were set through consultation with the instrument developers at the NOAA Chemical Sciences Laboratory.

With regard to flight orientation, we do not believe that this would cause significant issues. Under most flight conditions, the aircraft angle of attack and sideslip angles are less than 3 degrees. We do not believe that these small angles would cause significant deviations in the collection efficiency of the inlet system.

RC:   *P5, line 105 -106, It will be helpful to provide a summary of the measurement accuracy or uncertainty in this manuscript other than referring to the previous study.*

AR:   We understand the reviewers request, and the original version of this paper actually did have additional information on sensor and system accuracy. Unfortunately that version was returned to us by the journal for having too much text in common with previous publications. Therefore, that information was removed and is provided through references only.

RC:   *Table1.  It will be useful to include more information about the flight conditions, such as flight hours with SBF or the altitude range for the profiling flight. I think those have been included in the following sessions.*

AR:   We appreciate the desire of the reviewer to know more about the flight conditions experienced. Having said that, these conditions were variable enough, and user interests are broad enough, that it's not easy to provide a "one-size-fits-all" overview in a table. Additionally, the amount of work required to develop a climatology of different conditions experienced is significant enough that it extends beyond the scope of ESSD, and would require detailed analysis methods that are generally exceeding what should be included in a data paper. Having said this, we have included information on flight hours, flight times, and altitude ranges sampled in the manuscript (see figures 5 and 6, for example). Figure 5 covers all of the flights for the entire campaign, offering perspectives on how frequently each platform sampled each altitude.

RC:   *How many POPS flights do you have?*

AR:   Every RAAVEN flight carried the POPS instrument.  Wording was added to line 103 to clarify that all

RAAVEN flights were POPS flights. Of the completed flights, four did not produce any good POPS data due to overheating of the instrument. This information has been added to the manuscript.

**RC:** *P8, line 175, What is the accuracy of the derived quantities? What is the time resolution of the re-sampled data? 1 Hz or 10 Hz?*

AR: Derived quantities include the time_offset, Flight_Flag, Flight_State, altitude, fast air temperature, data quality flags, and wind variables. Information on estimated biases in the wind variables is included in Table 2, relative to radiosonde-derived winds, and the fast air temperature is calibrated in flight using the slow-response temperature sensor and the linear relationship between coldwire voltage and the temperature. We're not sure how to quantify accuracy for the other variables. The time resolution of all data in the B1 files is 10 Hz. This information has been added to the manuscript.

**RC:** *Table 2. What are the sources of errors for these measurements?*

AR: There are a handful of error sources, as with any observational platform. The RAAVEN and CopterSonde have different sensor payloads with slightly different sampling accuracies and biases, as described in de Boer et al., 2023. Per the comparison of each platform to radiosonde observations (Table 1 in the referenced article, partially reproduced below as Table 1 in this response), the mean differences between the CopterSonde and RAAVEN during TRACER lie within the mean biases in the previous study. The derivation of wind observations is also different, since the RAAVEN uses a combination of differential pressures from a multihole pressure probe to estimate angle of attack and sideslip and information from the onboard inertial navigation system to calculate winds. Therefore, it is more sensitive to low winds than the wind vane algorithm used to estimate winds onboard the CopterSonde. Furthermore, the sampling strategy of each platform is slightly different; the RAAVEN covers approximately a 200 m diameter area at each gridded height level when flying orbits and nearly 2 km of distance when covering racetracks, capturing information on temporal and spatial variability that the CopterSonde is not able to capture with its vertical flight mode. Also contributing to the differences illustrated here is the fact that there are likely small but persistent spatial gradients related to the surface properties at the operational sites. For example, at the BRZ site, the CopterSonde was operating downwind (in the case of sea breeze conditions) from a series of ponds. While the RAAVEN also had water features nearby, the upwind area during sea breeze conditions was drier, likely resulting in a different surface forcing locally. Additional text has been added to section 5 to help clarify some of these drivers of differences, and better explain the intent of the intercomparison effort.

| Test | CU RAAVEN | OU CopterSonde |
|---|---|---|
| T(K) | 0.31/0.17 | -0.12/0.23 |
| RH (%) | -1.62/1.37 | -4.06/1.93 |
| p (hPa) | -0.57/0.43 | -0.70/0.36 |
| q (g kg$^{-1}$) | -0.04/0.21 | -0.61/0.26 |
| Wind Speed (m s$^{-1}$) | 0.58/1.70 | -0.70/2.12 |
| Wind Direction (deg) | -3.76/6.45 | 1.15/13.55 |

Table 1: Mean and standard deviation of the bias values derived from comparison with radiosondes for each platform. The "total comps" row at the bottom is the number of comparison points (altitude bins) used in calculating these statistics. Table 1 from de Boer et al., 2023

**RC:** *Line 273 -274, What information can we gain from these 46 profiles?*

**AR:** The authors made a miscalculation, and there are a total of 44 profiles; this change is reflected in line 309. The 44 profiles make up all collocated profiles completed within 15 minutes of each other across the entire campaign. They are meant to show general agreement between the systems across the four months of the campaign, and provide potential data users with some reasonable assurance that the data can be used together in scientific analysis. A more comprehensive comparison between these systems and against traditional baloon and tower-mounted sensors is provided in de Boer et al., 2023.

**RC:** *Line 283 -285, Do you have the comparison of the vertical wind data?*

**AR:** No. At the moment, the CopterSonde wind algorithm can not derive vertical velocities, so there could be no intercomparison there. The vertical velocity data from the RAAVEN were compared to those from a tower-mounted sonic anemometer in de Boer et al., 2023. In that comparison, the aircraft was operating in close proximity to the tower. No such tower-based measurements were available during TRACER for comparison.

**RC:** *Line 294, How do you know it is due to the spatial difference, not the sensor uncertainty or discrepancy between sensors?*

**AR:** The sensor uncertainties for both platforms were added to Table 2 to add context to the contribution of sensor uncertainty to overall observational differences. Below, I have included an updated figure to include half of the flights from UHCC (where the aircraft were operating very close together) and the other half from BRZ (where the aircraft were operating up to 5 km apart), and it is visible the correlation is stronger at UHCC. Nevertheless, we have significantly reworded the intercomparison section to better describe the purpose of these intercomparison figures.

**RC:** *Appendix A1 needs to include more information and explain the variable names. For example, what is the POPS_LDM?*

**AR:** Unfortunately, adding this information into table A1 was not practical due to space considerations. However, as both reviewers brought up the need for additional information on POPS variables, we added additional text into section 4.1 to provide a more detailed overview of these variables.

**2. Reviewer 2**

**RC:** *- For the RAAVEN data there is much information about the quality flags which are really useful. The NetCDF files have relatively little meta-data information so that they are hard to use without the manuscript and additional information. For the POPS data, even the manuscript has very little information and does not explain the variables that are found in the dataset sufficiently clear. A more detailed description in Table A1 could help here.*

**AR:** We appreciate the reviewer's feedback on the data files. The NetCDF files contain information on units and a description of each variable. Additionally, they include the specific details of what individual flags mean and how they are defined. We would be curious to know what else the reviewer is hoping might be included in the metadata of the NetCDF files and what makes the data "hard to use". We agree that the manuscript could use additional information on the variables provided by the POPS. As such, we have added text to sections 2.1 and 4.1 to provide additional information on the POPS variables and the calculation of flow-corrected concentrations. Unfortunately, adding this information into table A1 was not practical due to

[Figure]

Figure 1: Updated Figure 8 to include more closely colocated flights from UHCC

space considerations.

**RC:** *- The coptersonde a0-data is not well explained, neither in the manuscript nor in the NetCDF-files themselves. Many variables cannot be interpreted by the user because they do not even have proper long names or units. It is thus questionable if this raw data should be made publicly available in that form. The c1-format can be well understood and used in contrast to that. It could maybe help to separate them in different directories if that is still feasible and make it more clear that a0-data is not meant for direct usage. If it is meant for public usage there should be a list of variables with explanation.*

**AR:** The a0 and c1 nomenclature is standard for US Department of Energy Atmospheric Radiation Measurement (ARM) datasets. We can ask to separate them in the ARM data repository using a folder structure. The a0 data are included to allow alternative post-processing, since there are various wind estimation algorithms. Following best practices, both the original data (a0) and subsequently processed data (c1) sets are archived in case others needed a different data resolution or to test wind vector algorithms. Lines 331-332 were added to tell the reader to contact the authors for more information regarding the a0 variables.

**RC:** *- There are no uncertainties specified for the measured variables. There is the comparison of the two*

*systems in Table 2, but the standard deviation for that will include errors due to atmospheric variability. It would be good to give estimates of uncertainty for the sensors and / or derived quantities, even if it is only the specified values by the sensor manufacturers. They could for example be included in Table A1 and A2. Clearly, uncertainty estimations for wind measurements with UAS are challenging, but have been done previously.*

AR: The manufacturer sensor uncertainties for temperature and relative humidity, and estimated wind biases were added to Table 2. The biases provided for wind speed and direction are biases calculated against ARM radiosondes in de Boer et al., 2023. This explanation was added to the caption for Table 2. The reader is directed to de Boer et al., 2023 numerous times throughout the manuscript if they are interested in additional information on system and sensor measurement accuracy and performance. Additionally, we have provided the reviewers an overview of the biases and standard deviations calculated relative to radiosondes in de Boer et al., 2023 as Table 1, above.

RC: *p.6, Fig.3: It is hard to read the maps. A scale would help to understand the dimensions better.*

AR: A scale for each of the smaller maps was added to help readability.

RC: *p.7, Fig.4: I like the overview of flights within the campaign period. I wonder if it could be expanded a little bit to include number of flights per day by each UAS and the flight patterns or main objectives, including IOPs. Maybe a table would be more suitable in that case or could be added. It would make it easier to navigate through the dataset.*

AR: The number of flights per day by each platform have been added to Figure 4. The flight patterns shown in Figure 6 are the primary flight patterns for the entire campaign for both platforms. The decision to fly the same flight patterns for every flight was made prior to the start of the campaign to allow for statistical comparisons. The only deviation from these flight patterns was when clouds or precipitation were nearby. Under these circumstances, RAAVEN flights may have been cut short or altered, and/or altitude limits may have been adjusted, but this did not happen often. The objectives were consistent throughout the campaign; we aimed to gather high-resolution ABL and aerosol data throughout sea breeze events, leading up to and after convection, through ABL transitions. There were no additional IOPs conducted by the UAS platforms, with all flights happening within the broader 4-month TRACER IOP period. The observations we could gather were highly dependent on the weather conditions. While a climatology of observations is a future step, it would require significant analysis and categorization to describe the flight conditions which is beyond the scope of an ESSD manuscript.

RC: *p.11, l.220: i guess the MHP is only calibrated for 20 degree AoS and AoA?*

AR: The probe is actually calibrated to +/- 15 degrees AoA and AoS. This is why values between 10 and 20 are flagged as suspect, as they are pushing the edges of the calibration. Beyond 20 is clearly not to be trusted and flagged as bad. We have added language to the manuscript to state that the probe is calibrated to +/- 15 degrees.

RC: *p.11, l.224: "below 5" please add units and explain the threshold.*

AR: The units ($m^2s^{-2}$) were added, and we corrected this text to state that values where the window variance were above (not below) were flagged as suspect. This value was found through manual review of this value relative to measured winds, and is meant to flag points that were potentially impacted by blockage of the MHP, or by other irregular flow situations.

RC: *p.11, l.225: what is the flight_flag?*

AR: The flight_flag is binary variable that is set at 1 if the aircraft is flying and 0 if the aircraft is not flying. This is described in lines 180-184. We have updated the text to label flight_flag as a binary variable.

RC: *p.11, ll.240ff: it is probably more clear to explain that the Flight_State is a binary code and explain the digits.*

AR: We have updated the text to read: These flags include the "Flight_Flag" introduced earlier in the manuscript, as well as a second "Flight_State" flag, which is a three-symbol binary variable. The Flight_State flag offers insight into whether RAAVEN is flying straight (0 in the ones place) or is turning (1 in the ones place), whether RAAVEN is descending (0 in tens place), level (1 in tens place), or ascending (2 in tens place), and whether RAAVEN is in flight (1 in hundreds place) or not (0 in hundreds place). For example, if a data user wanted to analyze straight, level flight legs, they would search for data with Flight_State equal to 110.

RC: *p.12, l.251: Is any filtering performed before downsampling? This could be important to avoid aliasing effects if you want to analyse data with regards to turbulence.*

AR: The low-pass FIR filter mentioned in line 258 is applied to the raw data prior to averaging and downsampling.

RC: *p.12, l.266f: when you say the data has been removed in calculations, this does not mean that they are flagged or removed in the c1-data I suppose!? I can find wind direction measurements for wind speeds <2m/s there.*

AR: The wind speed and direction data below 2 m/s remain in the files because they are not erroneous, just less reliable. For system intercomparison in this manuscript, the Coptersonde wind speed and direction data were filtered out below 2 m/s for a more representative comparison.

RC: *p.13, l.281: "within 6 m" is this correct? The drone is basically profiling right next to the lidar?*

AR: Yes, that is correct.

RC: *p.13, l.288: Where exactly was the "colocated" comparison, at BRZ or UHCC?*

AR: Perhaps "colocated" is too strong of a word (and we updated this word to co-located in the manuscript as it appears to be a more commonly used form). This comparison includes time periods where both aircraft were operating at BRZ or UHCC at the same time. Having said this, there was around 2500 m of separation between the aircraft at BRZ, and around 500 m at UHCC. We have updated the wording in the manuscript to reflect this reality.

RC: *p.13, ll.288ff and Fig. 7: The scatter plots here can be quite misleading, because they suggest quite a bad correlation between the two systems actually. I understand that it is meant to show the heterogeneity and the differences between the sites, but I doubt that it is the best way of presenting it. I would suggest a scatter plot for the colocated measurements at UHCC to show the good comparison between the two systems and some kind of presentation of the data from different locations in a separate graph. That could be showing temperature, humidity and wind on a map, or a direct comparison of the time series or mean values for the two measurement systems at different locations.*

AR: The scatter plots are meant to show general (not exact) agreement between the two platforms, and were not meant to provide any insight into potential differences between the different measurement sites. We have removed some of the language that implied that we were attempting to demonstrate consistent location-dependent differences using these scatter plots. We decided to alter Figure 7 to include all flights included in the comparison instead of selecting unique cases. We believe it shows the general agreement more completely. It is clear that there will be differences in the measurements resulting from the variability within

the atmospheric boundary layer, and the intent here was to include as many points as possible to attempt to reduce that relatively random variability from skewing any comparison. Having said all of this, a more thorough comparison is available in de Boer et al., 2023, where the two platforms were intentionally profiling very close to one another at the same time, and also compared to more traditional measurements systems (e.g., radiosondes, towers) to provide a more detailed comparison. We have directed readers to this reference for more information on the relative accuracies of the two platforms and have added language in the manuscript to clarify the intention of Fig 7.

**RC:** *p.15, Fig. 8: If I understand it correctly, the figure shows a mix of data from both sites. It is a bit hard to get much insight from the plots as they are. Maybe it would help to distinguish measurement sites or time of day within the plot. that could possibly be done by different shades of the used colors or multiple separate plots.*

AR: The reviewer is correct in thinking that both sites are represented in these figures. The primary goal of the figure is to show the range of conditions sampled across the whole campaign by the two platforms. The authors wanted to provide a simple snapshot of conditions, but not conduct any analysis of those conditions or investigate the differences between the two different sites, as such analysis would be beyond the scope of an ESSD article.

---

## Author Response (AR2)

**Authors' Response to Reviews of**

**Data collected using small uncrewed aircraft system during the TRacking Aerosol Convection Interactions ExpeRiment (TRACER)**

Francesca Lappin, Gijs de Boer, Petra Klein, Jonathan Hamilton, Michelle Spencer, Radiance Calmer, Antonio R. Segales, Michael Rhodes, Tyler M. Bell, Justin Buchli, Kelsey Britt, Elizabeth Asher, Isaac Medina, Brian Butterworth, Leia Otterstatter, Madison Ritsch, Bryony Puxley, Angelina Miller, Arianna Jordan, Ceu Gomez-Faulk, Elizabeth Smith, Steven Borenstein, Troy Thornberry, Brian Argrow, and Elizabeth Pillar-Little
*Earth System Science Data,* 2023-371
* * *
RC: *Reviewers' Comment*,     AR: Authors' Response,     ☐ Manuscript Text

**RC:**  *p.11, l.237ff: Calibration of MHPs is usually done with high-order polynomials. An extrapolation outside the calibration range can be very high and does not have traceable uncertainty and thus should be discarded.*

AR:  We appreciate the reviewer's understanding of the MHP calibration process and this comment. We agree that extrapolation beyond the calibration range is dangerous and should be treated with extreme caution. However, rather than discarding data, we prefer to use data quality flags. As stated in the manuscript, any data points where the absolute values of alpha or beta were greater than 10 degrees were flagged as "questionable". Further, any data points where alpha or beta are deemed to be greater than 20 degrees are marked as "bad". The primary recommendation for data users is to only use points where data are flagged as "good", i.e., where alpha and beta are well within the calibrated range. To make this clear, we have added the following text to the manuscript (l.245-248): "Data users should be aware that extending multihole probe calibration coefficients beyond angles tested in the wind tunnel can result in highly non-linear errors in estimation of angle of attack and sideslip angle. Such errors significantly impact wind estimation. As such users are advised to only use wind data where the wind_flag variable is equal to zero. These values are well within the calibrated range of the multihole probe."